# Middle through late Holocene long-distance transport of exotic shell personal adornments in Central West Patagonia (southern South America). The archaeomalacological assemblage of Baño Nuevo 1

Heidi Hammond[1], Leandro Zilio[1], Amalia Nuevo-Delaunay[2], César Méndez[3]*

1 Consejo Nacional de Investigaciones Científicas y Técnicas (CONICET), Universidad Nacional de la Patagonia San Juan Bosco, Esquel, Argentina, 2 Centro de Investigación en Ecosistemas de la Patagonia (CIEP), Coyhaique, Chile, 3 Estudios Aplicados, Escuela de Antropología, Pontificia Universidad Católica de Chile, Santiago, Chile

* cesar.mendezm@uc.cl

## Abstract

The exchange of information and social interactions on broad spatial scales between human groups in the past can be studied through the provenance of key indicators of distant origin recorded at archaeological sites. The remains of shells of mollusk species, especially when crafted as elements of personal ornaments, express aspects of the behaviors and valuations for the populations that selected, transformed, and exchanged such items. In the southern cone of South America, past hunter-gatherer groups traveled long distances and interacted with communities distributed throughout the territory to acquire goods for technological use, visual display or considered highly valued materials. When recorded at distant locations, these goods of extra local origin are very informative regarding the differences between commonly used home ranges and the occasional access to remote spaces. We present the results of the analysis of the archaeomalacological assemblage of the Baño Nuevo 1 site, a cave with exceptional preservation conditions in Central West Patagonia. This site has yielded a diverse group of artifacts made of shells with origins from multiple distances, as well as evidence of the use of marine, freshwater, and terrestrial species. Its deposits, which extend over the last 11,000 years, reveal an antiquity of at least the middle Holocene for the acquisition, manufacture, use and transport of goods as personal ornaments from shells in the macroregion.

## Introduction

The Baño Nuevo 1 cave, located in Central West Patagonia (southernmost South America), is a renowned archaeological context in the region because of the extent of its deposits, with paleontological and archaeological evidence across the last 16,000 years [1, 2]. The cave is a

**Data Availability Statement:** All relevant data are within the manuscript.

**Funding:** Funded by the Agencia Nacional de Investigación y Desarrollo (https://anid.cl) through the ANID FONDECYT 1210042 (CM) and ANID Regional R20F0002 (AN-D) grants. The funders had no role in study design, data collection and analysis, decision to publish, or preparation of the manuscript.

**Competing interests:** The authors have declared that no competing interests exist.

volcanic bubble that, due to its shape and small entrance, allowed the survival of a very rich material record, among which is a group of ten individuals dated to *ca.* 10,200 calibrated years before present (cal BP), corresponding to the earliest directly dated bioanthropological assemblage in Patagonia and one of the oldest in South America [3, 4]. However, at the most recent levels (upper), the context contains a varied assemblage of materials, mainly organic, which are not frequent in other sites in the region [5, 6]. Baño Nuevo 1 is located 100 km from the closest point to the Pacific Ocean and 320 km from the closest point to the Atlantic (Fig 1). Despite these distances, among the remains recovered in the excavations, there is a group of objects of personal adornment manufactured with the exoskeletons of mollusks, both marine and terrestrial, as well as other archaeomalacological fragments without evidence of anthropogenic modification.

In this paper, we present the results of the analysis of the archaeomalacological assemblage recovered at Baño Nuevo 1 and discuss their implications as exotic materials. The importance of analyzing this type of evidence lies in the fact that mollusks play a fundamental role in the interpretations of transport and human circulation because their biogeographic habitats allow us to determine the probable areas of collection, such as, for example, marine coasts. Many species of mollusks were part of the diets of human populations in the past, and in some cases, they were intensively used and consumed in different areas of the world, as they constitute fixed and predictable resources [7]. The discovery of mollusks at archaeological sites distant from their catchment areas allows us to discuss them as circulation indicators in the past [8, 9]. In South America, large networks of exchange of *Spondylus* shells occurring between the coast and up to elevations above 3,000 m above sea level, starting at 4,600 cal BP, have been described [10]. These networks involving shells have deep chronological roots as indicated by evidence in northern Chile dating back to the end Pleistocene [11]. Particularly, personal ornaments made from mollusks of nonlocal or exotic origin must be understood within the framework of interaction, communication, and information circulation networks [12, 13]. Often, these types of object can be exchanged or offered as gifts to members of other distant groups, functioning as goods to reinforce the establishment and maintenance of a network of social relationships within the framework of population movements not related to subsistence activities, that is, as "nonutilitarian" goods [13].

In Patagonia, obsidian -among other materials- is routinely used as an exotic element to understand the ranges of action and procurement mechanisms of the past hunter-gatherers that inhabited the region [14, 15]. The role of goods of coastal origin has also been explored to evaluate the magnitude of the home ranges [16]. Thus, coastal resources have been considered not only as a food but also as raw materials for the manufacture of utilitarian objects and personal adornments. In Patagonia, hanging-adornments made from the exoskeletons of mollusks have been recorded both in residential contexts and in burial sites associated with the interred individuals [17–23]. However, there is no direct ethnographical evidence of the management of shell mollusk exotics in inland locations, thus underscoring the role played by the archaeological data in addressing this issue. Inland societies of Central and South Patagonia developed an strategy to avoid contact with the western world during the nineteenth and twentieth centuries (and likely earlier), which resulted in infrequent ethnographic accounts and even occasional areas with no reported indigenous populations [24–26].

In this article, we investigate the malacological evidence at Baño Nuevo 1 to better understand the long-distance transportation of exotic goods. The probable sources of origin of the different species illustrate the breadth of the interaction spaces of the hunter-gatherer populations. Together, the probable uses of the malacological remains are analyzed and evaluated. Finally, the chronology of the human occupation of Baño Nuevo 1 allows us to discuss the antiquity of the managing and transportation of ornamental malacological goods in Patagonia.

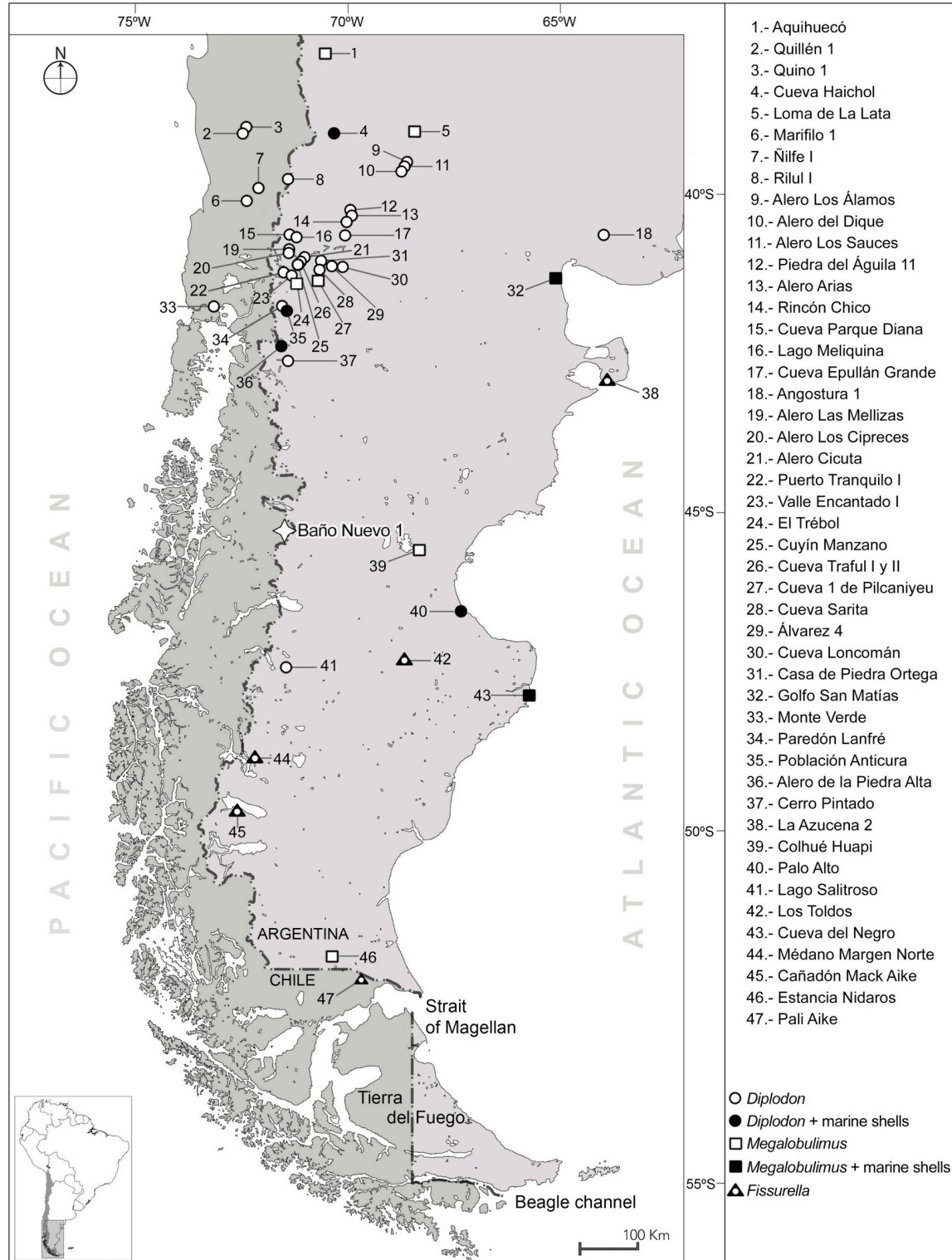

1.- Aquihuecó
2.- Quillén 1
3.- Quino 1
4.- Cueva Haichol
5.- Loma de La Lata
6.- Marifilo 1
7.- Ñilfe I
8.- Rilul I
9.- Alero Los Álamos
10.- Alero del Dique
11.- Alero Los Sauces
12.- Piedra del Águila 11
13.- Alero Arias
14.- Rincón Chico
15.- Cueva Parque Diana
16.- Lago Meliquina
17.- Cueva Epullán Grande
18.- Angostura 1
19.- Alero Las Mellizas
20.- Alero Los Cipreces
21.- Alero Cicuta
22.- Puerto Tranquilo I
23.- Valle Encantado I
24.- El Trébol
25.- Cuyín Manzano
26.- Cueva Traful I y II
27.- Cueva 1 de Pilcaniyeu
28.- Cueva Sarita
29.- Álvarez 4
30.- Cueva Loncomán
31.- Casa de Piedra Ortega
32.- Golfo San Matías
33.- Monte Verde
34.- Paredón Lanfré
35.- Población Anticura
36.- Alero de la Piedra Alta
37.- Cerro Pintado
38.- La Azucena 2
39.- Colhué Huapi
40.- Palo Alto
41.- Lago Salitroso
42.- Los Toldos
43.- Cueva del Negro
44.- Médano Margen Norte
45.- Cañadón Mack Aike
46.- Estancia Nidaros
47.- Pali Aike

○ *Diplodon*
● *Diplodon* + marine shells
□ *Megalobulimus*
■ *Megalobulimus* + marine shells
△ *Fissurella*

**Fig 1. Distribution of archaeological sites with malacological evidence in Patagonia showing the occurrence of the main taxa identified in Baño Nuevo 1.** Vector map data: Natural Earth (public domain at http://www.naturalearthdata.com). Figure produced using QGIS 3.22 and Inkscape 1.2 software.

## The archaeological context of Baño Nuevo 1

Central West Patagonia is located along the Andes of the southern cone of South America. This mountain range produces a west-east decrease in precipitation as a result of the forced subsidence of the westerly winds [27]. The Andean valleys cross a forested gradient that culminates in a semiarid environment of open steppes to the east of the region [28]. The Baño Nuevo 1 cave is located in one of these valleys, that of the Ñirehuao River, which was shaped by glacial lobes for at least the last 150,000 years [29]. At approximately 19,000 cal BP, the last glacial retreat began, a process associated with cold and dry conditions that persisted until 14,800 cal BP, as detected in the fossil pollen records obtained from lakes in the region [30–32]. The increase in effective humidity indicated by a rise in tree pollen taxa frequencies, and associated with warmer conditions, occurred as a stepwise process that culminated at 8,000 cal BP, when the maximum expansion of the forests to the east was detected [33, 34]. During the Holocene, the open steppe environment was characterized by variations in the effective humidity that allowed the development and contraction of the forests at their margins, with a minimum observed at 3,000 cal BP, and the later development of conditions similar to those currently observed [31].

The studied cave is located at the foot of a basaltic butte of the Baño Nuevo Volcanic Complex (ca. 122 Ma) that stands out among the fluvio-glacial deposits at the bottom of the valley [35]. It was initially excavated in 1972 by Luis Felipe Bate and then, from the 1990s onward, by a team led by Francisco Mena [5, 36]. There are two distinct areas within the cave: the central part and the rear are characterized by fine sediment particles deposited under a low energy regime, while the entrance is covered by clast-supported colluvial deposits that entered the cave from the outside. Excavations focused on the center and the rear. The stratigraphic sequence is defined by 49 radiocarbon ages, of which 35 were obtained from postglacial levels that serve to characterize the chronology of human presence [6]. Sedimentation inside the cave was relatively continuous without major erosive disconformities [37]. It began with a stratigraphic unit (SU) of compact coarse sand with horizontal lamination, which suggests an origin in an aquatic environment. Two silty units lay above it: SU5, formed between 13,000 and 16,200 cal BP, and SU4, with ages between 10,900 and 13,000 cal BP. The last SU includes three subunits that are distributed discontinuously and are recognized for having different degrees of compaction, types of inclusions and proportions of highly decomposed plant material. A greater presence of fractured rocks in the deposit suggests alternating cycles of humidity and dryness during the transition between the Pleistocene and the Holocene [37]. SU3 consists of fine silts and extends between 4,300 and 10,900 cal BP. It includes five minor stratigraphic units, also horizontally discontinuous, that like SU4, vary in terms of compaction, inclusions, and plant material. This layer has a very good preservation, as it includes abundant bones, feathers, hairs, vegetables, charcoal fragments, and twigs, among other materials. This SU includes most of the human evidence, hence it is also the one with the greatest alteration processes, primarily because of the burials of at least 10 individuals arranged mainly on the margins and rear of the cave [3]. On top of this SU, there is SU2, corresponding to clayey silts with a significant proportion of clasts as a result of the disintegration of the rock walls of the cave. This formed between 2,900 cal BP and the beginning of the 20th century and occurs under a unit of sheep dung (SU1), whose antiquity dates to the establishment of the Baño Nuevo ranch.

A geoarchaeological study suggests that the beginning of the deposit, at 16,200 cal BP, was in the context of ephemeral accumulations of water, which entered the cave due to the topography of the valley [37]. These results are consistent with the evidence and timing of the retreat of a proglacial lake from the valley [30]. SU4 and 5 contain evidence of various extinct fauna

taxa whose direct ages indicate that they inhabited or were deposited in the cave prior to human occupation [38, 39]. The biological activity inside the cave increased at the beginning of the Holocene; this activity is attributed not only to the presence of humans (including domestic and funeral activities) but also to fauna, which alternated with human occupations [37]. Three components have been defined in order to study the occupation of the cave: early (10,900–8,900 cal BP), mid (8,900–5,700 cal BP) and late (5,700–2,900 cal BP) [40]. However, a more detailed analysis of the radiocarbon ages for the site and their distribution suggests that the occupation of the cave was not continuous throughout these components but rather a series of brief events (N = 15) interspersed with prolonged periods without occupation [41].

## Materials and methods

The collections considered in this research are curated at the Museo Regional de Aysén (Coyhaique, Chile) and were studied with the proper institutional authorization. The analyzed malacological assemblage comes from different excavation units in the Baño Nuevo 1 cave. Excavations were performed in 1996 and between 2004–2006 and the senior scientist responsible for them granted permission to analyze the archaeological material presented in this article. The authors carried out no field work for this study. The materials were mainly recovered from SU2, 3, and 4 (Table 1). Given that the remains are generally small (usually <1 cm) and

**Table 1. Archaeomalacological materials recovered at the Baño Nuevo 1 site.**

| Specimen | Taxa | Conservation | Type | Unit | SU |
|---|---|---|---|---|---|
| 1 | *Megalobulimus lorentzianus* | Complete | Pendant | 9D | 2 |
| 2 | *Megalobulimus lorentzianus* | Complete | Pendant | 8C | 2 |
| 3 | *Megalobulimus* sp. | Fragment | Pendant | ND | ND |
| 4 | Veneridae | Complete | Bead | 9D | 3A |
| 5 | Undetermined | Complete | Bead | 9D | 2A |
| 6 | Undetermined | Complete | Bead | 7C | 3 |
| 7 | Undetermined | Complete | Bead | N/D | 3 |
| 8 | Undetermined | Complete | Bead | 9D | 3A |
| 9 | Undetermined | Complete | Preform | 8C | 2 |
| 10 | *Fissurella* sp. | Complete | Bead | N/D | Surface |
| 11 | *Fissurella* sp. | Complete | Bead | 2A | 1–2 |
| 12 | *Fissurella* sp. | Complete | Bead | 1B | 4 |
| 13 | *Fissurella* sp. | Complete | Bead | Rear sector* | ND |
| 14 | *Fissurella* sp. | Complete | Bead | 2A (E) | 3 |
| 15 | *Fissurella* sp. | Complete | Bead | 2A (E) | 3 |
| 16 | *Aulacomya atra* | 2 fragments | Unmodified | 9D | 3B |
| 17 | *Diplodon* sp. | 14 fragments | Unmodified | 9D | 4 |
| 18 | *Diplodon* sp. | 1 fragment | Unmodified | 9D | 4 |
| 19 | *Diplodon* sp. | 1 fragment | Unmodified | 9E | 3 |
| 20 | *Diplodon* sp. | 4 fragments | Unmodified | 9E | 3 |
| 21 | *Diplodon* sp. | 1 fragment | Unmodified | 9D | 3A |
| 22 | *Diplodon* sp. | 1 fragment | Unmodified | 6C | 3 |
| 23 | *Diplodon* sp. | 1 fragment | Unmodified | 8D | 2 |
| 24 | Undetermined | 1 fragment | Unmodified | 6B ext. 6A | 3A |
| 25 | Undetermined | 1 fragment | Unmodified | 2B (NW) | 4 |

* Material sieved from previous excavations. ND: no data.

the cave sediments are not very compact, it is not surprising that some specimens (N = 4; one case are fragmented pieces likely belonging to the same specimen), including one bead, were recorded in SU4, which dates mainly prior to human occupation. The vertical migration of small remains has been studied at the site previously, having detected the upward migration of dermal ossicles of *Mylodon* sp., a Pleistocene sloth, which were recorded over more recent human-made features, such as hearths [42].

The archaeomalacological materials were observed with a binocular magnifying glass (Zeiss Stemi DV4) and under a digital microscope (Dino lite AM4113ZT) to determine the species, genus, or family of the mollusks and for observing surface characteristics and distinctive traces. Bibliographic sources were used for the anatomical and taxonomic identification [43–53], and mollusk catalogs [54] and comparative collections of *Diplodon* sp. and of *Megalobulimus* shells of the species *M. lorentzianus*, *M. sanctipauli* and *M. abbreviatus* were donated by the Instituto de Biología Subtropical (IBS-Ma; CONICET—Universidad Nacional de Misiones). The identification process was based on the anatomical and taxonomic recognition of the exoskeletons of mollusks. In the archaeological remains, because only the hard parts of these organisms are preserved, identification was carried out considering the biogeographic distribution of the invertebrates and the distinctive characteristics of the exoskeletons, such as morphology, color, sculpture (ornamentation: relief pattern of the shell surface), shell contour, apex arrangement, characteristics of the peristome, and the aperture, among other features [55–58]. The dimensions of remains were measured with a gauge and recorded.

Evidence of intentional modifications, such as marks, perforations, polishing and incisions, was analyzed, as were taphonomic variables, such as color preservation, drying lines, signs of abrasion, presence of fissures, cracks and/or fractures, exfoliation, bioerosion, root marks, presence of carbonates, dissolution and heat alteration [20, 56, 59–61]. Ornate-hanging objects have been characterized as those artifacts of varied shape, generally small and durable, that are located on the body by some form of suspension, mainly perforations or notches, or they can also be objects that are sewn or fastened by means of a string to the body, clothing or other support or personal object [62]. The function of these pieces can be diverse, depending on the person who uses them but also on the way in which they are worn [61]. The modified artifacts were classified according to their morphology and elaboration stage into beads, pendants, and preforms. The beads correspond to objects with a central perforation and would have been used grouped with others alike [63–65]. Pendants have one or more perforations, which are frequently displaced with respect to the center in order to be fastened individually [20, 60, 63–65]. In some cases, the possible techniques for suspending such objects and trace evidence caused by their use were also evaluated [61]. Finally, we identified possible procurement areas where the hunter-gatherer groups could have obtained the raw materials used for manufacturing the ornaments, in addition to identifying the origin of the unmodified exoskeletons recorded.

## Results

From the analysis of the archaeomalacological remains recovered at the Baño Nuevo 1 site, an assemblage of 15 manufactured artifacts, i.e., beads, pendants and a possible ornamental object preform were identified (Fig 2 and Table 2), in addition to 10 fragments of exoskeletons of marine and freshwater mollusks without traces of anthropic modification (Table 1). Some of the remains, due to the absence of diagnostic parts or features, could not be identified at the family, genus, or species level. None of the manufactured artifacts (i.e., beads and pendants) were recorded in association with the skeletal material of the individuals buried in the site, hence there is no association of these materials with specific individuals.

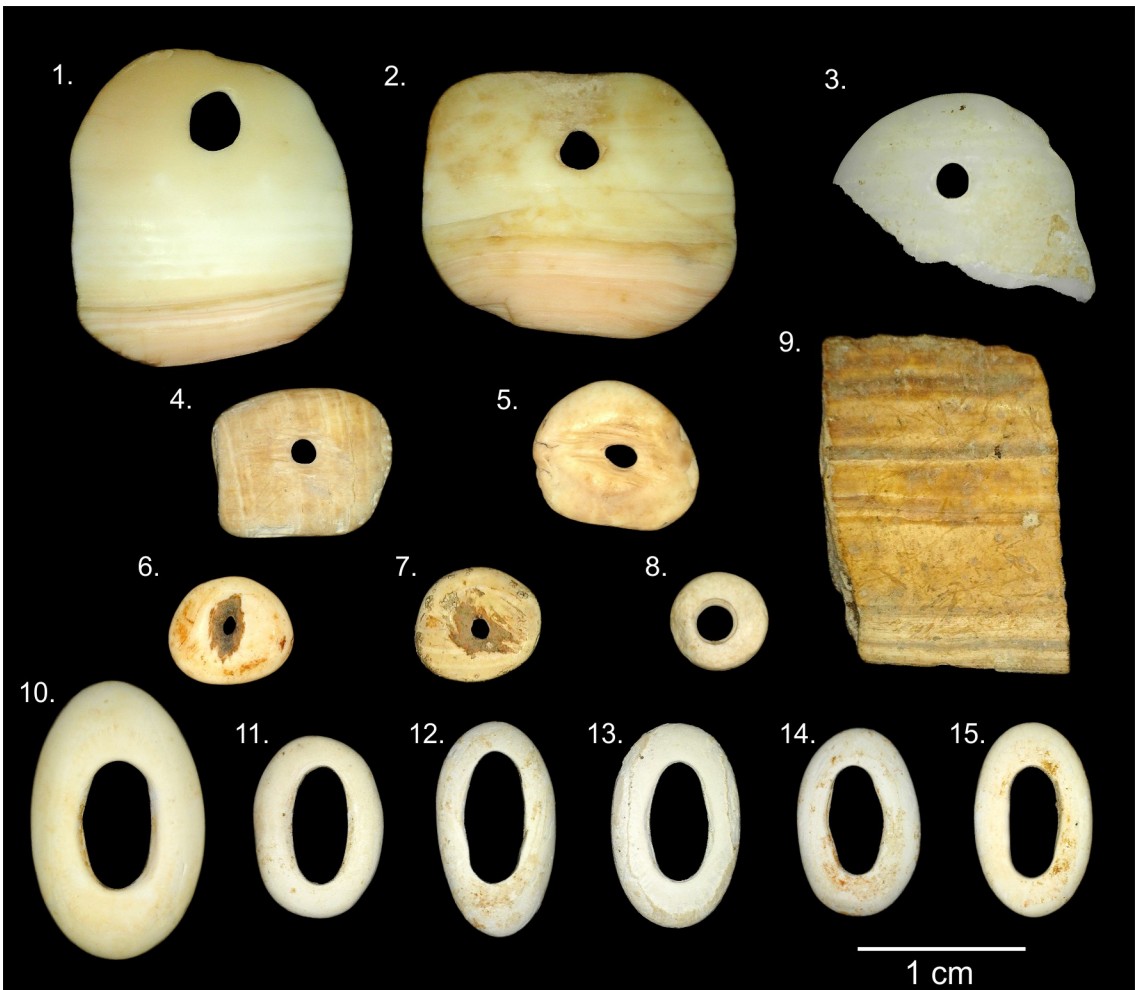

**Fig 2. Archaeomalacological artifacts from Baño Nuevo 1.** The numbering of the specimens is maintained throughout the text.

Specimens # 1 and 2 are complete pendants (Figs 2 and 3). Based on diagnostic features such as the presence of a characteristic outer lip or peristome of a thick, pink color, the ivory-colored shell, and thick, parallel and irregular ribs, we conclude these adornments were manufactured from a species of land snail, *Megalobulimus lorentzianus* (Doering, 1876) [49, 50]. Within these artifacts, part of the peristome and the aperture are preserved. Also, the original color of the shell is partially preserved. Both artifacts present a conical perforation made through the action of rotation and pressure with an instrument from the inner face of the shell. For specimen #1, the perforation has a smooth and highly polished surface; however, there are fine concentric incisions on the profile of the perforation, parallel in the hole, suggesting that it was created through rotational wear and pressure. Both the upper edge of the perforation and the nearest outer edge have a slight notch. In addition to these features, the internal face also shows wear, possibly generated using a string through the hole (Fig 3A). Both pieces are polished, a process that created smoothed contours, a shiny surface, and very fine striations. For specimen #2, the perforation is smooth, with polished surface contours. In turn, there are fine parallel incisions in the contour of the hole generated by the manufacturing process. Both the upper edge of the perforation and the nearest outer edge have a slight notch (Fig 3B). This piece could have been tied or sewn, similar to specimen #1.

**Table 2. Characteristics of the modified archaeomalacological specimens at the Baño Nuevo 1 site.**

| Specimen | Taxa | Type | Morphology | Dimensions (mm) | | | Perforation (mm) | |
|---|---|---|---|---|---|---|---|---|
| | | | | Long | Width | Thickness | Diameter max./min. | Kind |
| 1 | *Megalobulimus lorentzianus* | Pendant | Subquadrangular, curved | 18.7 | 16.9 | 1.2 | 3.5/2.9 | Conical |
| 2 | *Megalobulimus lorentzianus* | Pendant | Subquadrangular, curved | 18.6 | 15.6 | 1.4 | 2.3/2.2 | Conical |
| 3 | *Megalobulimus* sp. | Pendant | Subquadrangular, curved | 14.8 | 11 | 1.9 | 1.4/1.3 | Conical |
| 4 | Veneridae | Bead | Subquadrangular, flat | 10.6 | 9.1 | 1.8 | 1.6 | Biconical |
| 5 | Undetermined | Bead | Semicircular, flat | 10 | 8.7 | 2 | 2.4/1.6 | Biconical |
| 6 | Undetermined | Bead | Circular, flat | 7.7 | 7 | 1.6 | 0.5 | Biconical |
| 7 | Undetermined | Bead | Semicircular, flat | 7.4 | 6.3 | 1.4 | 1.6/1.3 | Biconical |
| 8 | Undetermined | Bead | Circular, flat | 5.8 | 5.7 | 1.5 | 2.5 | Cylindrical |
| 9 | Undetermined | Preform | Rectangular, flat | 19.4 | 14 | 3 | ND | ND |
| 10 | *Fissurella* sp. | Bead | Oval, flat | 16.6 | 10.4 | 3 | 6.8/3.3 | Oval * |
| 11 | *Fissurella* sp. | Bead | Oval, flat | 10.6 | 7.6 | 1.7 | 6.1/2.5 | Oval * |
| 12 | *Fissurella* sp. | Bead | Oval, flat | 12.1 | 7.5 | 1.6 | 6.3/2.5 | Oval * |
| 13 | *Fissurella* sp. | Bead | Oval, flat | 11.5 | 7 | 1.7 | 5.8/2.2 | Oval * |
| 14 | *Fissurella* sp. | Bead | Oval, flat | 13.7 | 7 | 1.6 | 7/2.5 | Oval * |
| 15 | *Fissurella* sp. | Bead | Oval, flat | 11.1 | 7.2 | 1.3 | 5.5/2 | Oval * |

* Aperture or natural opening of the shell. ND: no data.

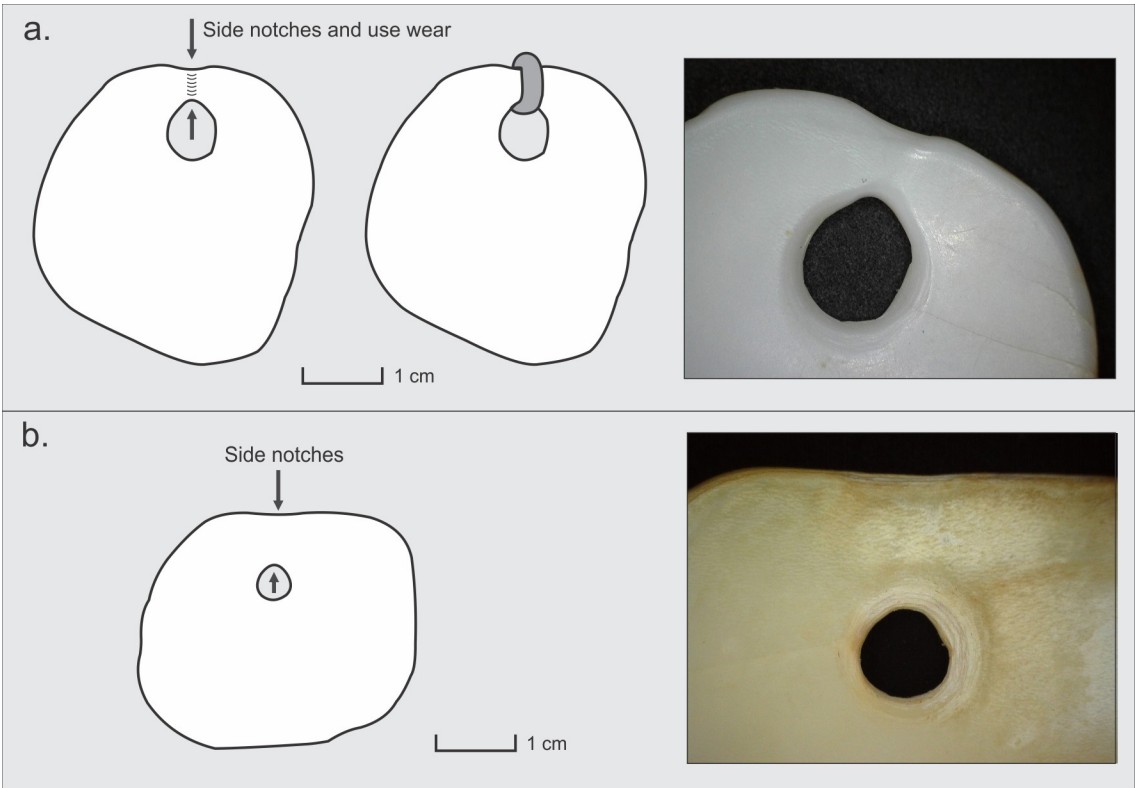

**Fig 3. Schematic representations of perforations and use traces (detailed photographs) Baño Nuevo 1 pendants.** a. specimen #1 and b. specimen #2.

Specimen #3 is a proximal fragment of a pendant (Fig 1) with a medial fracture and a single conical circular perforation; the largest opening located on the inside. Although the piece is polished, with shiny edges and faces, the original gastropod sculpture of parallel ivory-colored ribs, is preserved. The observed data indicate that a shell of the genus *Megalobulimus* sp. was used to craft this pendant.

Another of the hanging ornaments (specimen #4) is a light brown square bead (Fig 2). Both sides and edges are polished, creating a shiny surface and fine striations. On its external face, despite polishing, there are vestiges of the original sculpture of the shell, characterized by fine parallel lines. This sculpture is similar to bivalve specimens of the Veneridae family (clam). In the central sector of the bead, both faces are hollowed-out, thus reducing the thickness of the piece where the circular perforation is located. These hollowed-out areas have parallel incisions, indicative of scraping.

In addition, another 4 beads were recovered (specimens #5–8); however, due to the absence of diagnostic features and the advanced level of polishing, it was not possible to determine which mollusks served as the raw materials. Three of the beads have a circular to semicircular morphology (specimens #6–8), and in the central area, both faces are hollowed-out, thus reducing the thickness of the pieces at the oval perforation, of biconical type. These hollowed-out areas have parallel incisions as a result of scraping. Scraping was performed by applying direct pressure to the shell, possibly with a lithic tool. Scraping has been documented as a strategy to prepare the surface of an object with considerable thickness to be drilled [61]. The surfaces of the pieces are highly polished, with rounded edges. In addition, there are very fine marks as a result of this process, as well as a shiny surface. On both sides of beads #6 and 7 (Fig 2), there is presence of red sediment, probably ochre, on the central hollowed-out surface. In addition, in specimen #6, small carbonates are adhered, some on the intentionally polished surfaces, indicating that they likely formed after deposition. Finally, a small circular bead with flat faces and edges was identified (specimen #8); this bead shows evidence of intense modification and a cylindrical-type center perforation (Fig 2).

Another of the modified pieces is a fragment of the edge of a mollusk shell (specimen #9) whose taxonomy could not be defined (Fig 2). The external face has parallel light brown bands interspersed by other thinner dark brown bands. The object is nearly smooth. The fragment has smooth edges between 45° and 90°. It has a rectangular morphology that suggests that it could be a preform of an ornamental object [20]. This piece was modified at its edges using a cutting technique through the application of bidirectional movements with an instrument until generating a straight incision that allowed detachment of the desired fragment by pressure. In this case, the observed result is a preform with a subrectangular morphology. No other manufacturing-related traces are evident, such as abrasion, polishing of faces or edges, or drilling. The external and internal faces are the natural surfaces of the shell.

Finally, among the artifacts, 6 complete beads with an oval/elliptical morphology were recorded. These specimens were manufactured from keyhole limpets—*Fissurella* sp. -, using the natural aperture at the apex of the gastropod shell (Fig 2). Specimens #10 and 11 have a shiny surface, and specimens #12–15 have a dusty or porous surface with very little surface gloss.

Among the malacological remains without anthropic modifications, two shell edge fragments of the species *Aulacomya atra* were identified (Table 1). The external face has a characteristic purplish color with radial ribs to which the grooves or growth rings are superimposed in a perpendicular and concentric way. The internal face of the fragments is pearly and shiny.

Within the archaeomalacological assemblage, 23 shell fragments of *Diplodon* sp. (freshwater clam) were observed (Table 1). Among these, 4 edge fragments, 3 hinges and 2 right umbos were recognized. Most of these fragments preserve remains of the periostracum of brown

color, with an iridescent pearl-like color on the inside. A fragment of a right shell shows evidence of thermal alteration, which generated an opaque light brown color and eliminated the periostracum. Despite this, the characteristic iridescent pearl-like color of this mollusk was preserved on the inside.

Only two unmodified mollusk fragments could not be taxonomically identified: one is an edge fragment, with an iridescent pearl-like color on its internal face, and the other is white with a powdery or porous surface.

## Discussion

The archaeomalacological assemblage of the Baño Nuevo 1 site includes ornate pendant-type artifacts, beads, and a preform of an ornamental object, as well as fragments of exoskeletons of unmodified mollusks. In general, all the specimens are well preserved. Some of the remains were identified at the species level, and some were identified at the genus or family level. For some of the modified pieces, a high degree of polishing, and a lack of diagnostic features, sculpture and ornamentation precluded the identification of the mollusk species used as raw materials. However, from the taxonomic identification of some specimens, it was possible to recognize various mollusks of regional origin, as well as other extra-Patagonian mollusks. Marine, fluvial, and terrestrial species were recognized, which give the assemblage a great malacological richness. The different origins of the mollusks from Baño Nuevo 1 allow us to discuss the regional and extraregional transport of personal adornments and other unmodified objects. The artifacts were distributed chronologically from the beginning of the Holocene (top of SU 4) until 2,900 cal BP (SU2), when the site was abandoned. This implies the establishment of communication networks and the circulation of information and contact between different groups and distant territories. Next, the different types of archaeomalacological evidence from Baño Nuevo 1 are discussed.

### The remains of the local freshwater clam *Diplodon* sp.

In the analyzed collection, numerous unmodified exoskeleton remains of *Diplodon* sp. were identified. *Diplodon chilensis* (Gray 1828) is a clam from in-land fresh waters. They are oblong-elongated bivalves that can reach lengths of up to 85 mm [66]. Its shells have a thin periostracum with a glossy brown color; internally, it is pearly with a deep cavity for the anterior retractor muscle [46]. Some authors differentiate two subspecies: *D. chilensis chilensis* and *D. chilensis patagonicus* [67]. The first occupies the western area of the Andes, while the second occupies the eastern area. *D. chilensis patagonicus* currently inhabits the sandy and muddy substrates of mountain lakes, from the province of Mendoza to the vicinity of the Futalaufquén Lake, in Chubut (Argentina) [67] (Fig 4). This species is common in lakes in Central West Patagonia (Aisén, Chile). It is characterized by a relatively thin and fragile shell, which is why, in general, complete specimens at archaeological sites are scarce [22].

The presence of the *Diplodon* genus at archaeological sites in Patagonia dates to the early Holocene. Jackson and Jackson conducted a review of the occurrence of this mollusk at archaeological sites in Chile [46]. The earliest evidence is at the Monte Verde II site (42° S; Chilean Lake District), dated at 14,600 cal BP, north of Patagonia [68]. Further north, at the Marifilo 1 site (39° S), with occupations from the early Holocene to present times, there was a large number of *Diplodon* sp. specimens. This site had several occupational events, and this freshwater mollusk was a common resource in the subsistence of human groups [69]. At the Población Anticura site in the Río Negro province (Argentina), a fragment of an adornment made on *Diplodon* sp. was recorded in early Holocene deposits [20]. The fragments of *Diplodon* sp. at Baño Nuevo 1 were recorded in SUs with an associated chronology extending from

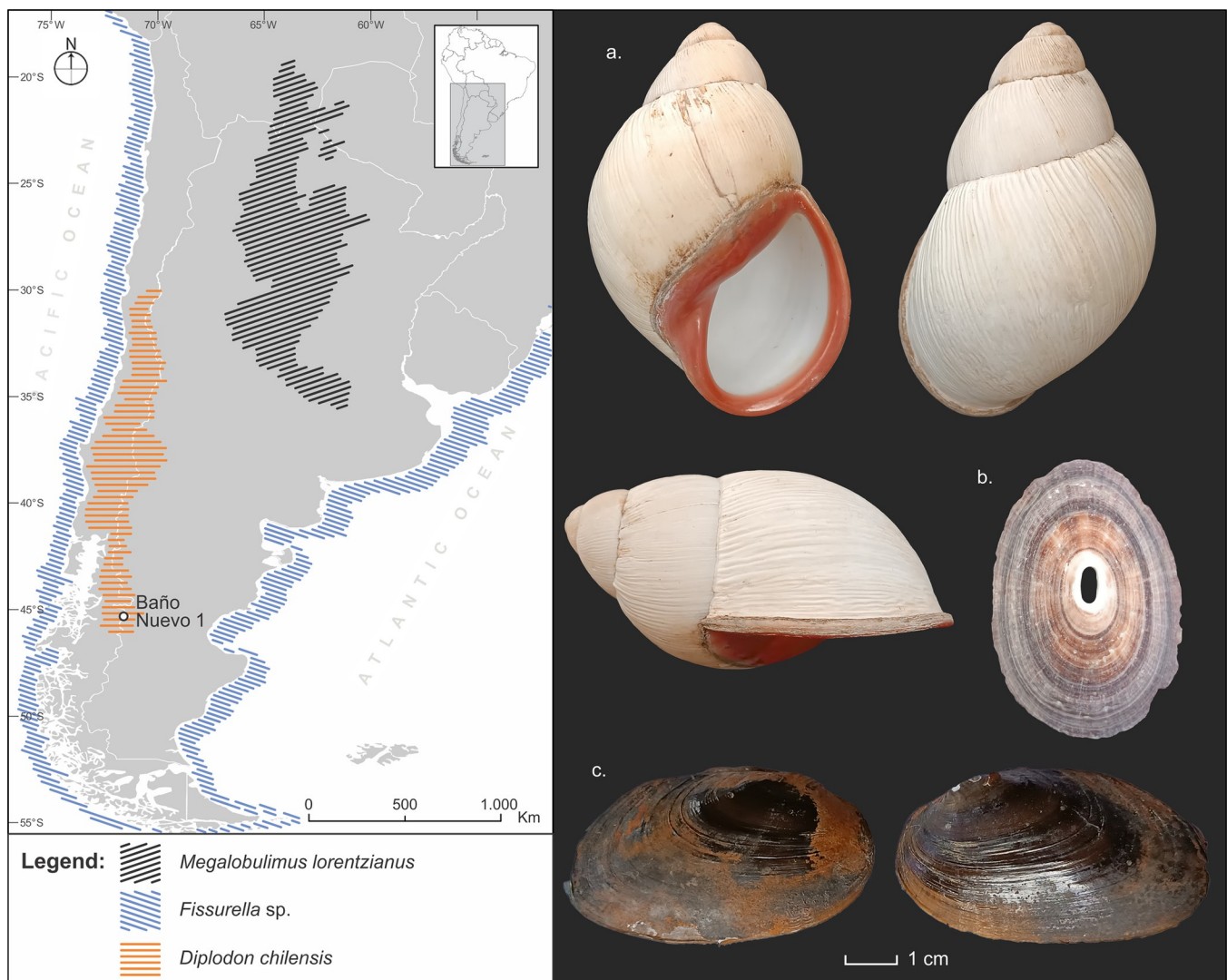

**Fig 4. Biogeographic distribution of mollusk shells from Baño Nuevo 1.** a. *Megalobulimus lorentzianus* [49, 50], b. *Fissurella* [43] and c. *Diplodon chilensis* [67]. These shells are part of the reference material used in this work. *Fissurella pulchra* is a representative example of the genus. Vector map data: Natural Earth (public domain at http://www.naturalearthdata.com). Figure produced using QGIS 3.22 and Inkscape 1.2 software.

11,000 to 2,900 cal BP; several of the artifacts, because they were located in SU4, are probably among the earliest anthropogenic evidence of occupation in the cave. The fact that most specimens were recovered from contiguous excavation units (especially 9D and 9E) in the southern part of the cave suggests a possible common origin for some of them.

Many of the *Diplodon* fragments from Baño Nuevo 1 preserve part of the periostracum. The periostracum is a protein layer that rapidly degrades when exposed to environmental conditions, leaving the structure of the calcium carbonate of the shell exposed to physical weathering and dissolution agents [70]. When exposed to environmental conditions, this membrane dries quickly, fractures and peels easily. The presence of the periostracum in shells at archaeological sites is often interpreted as an indicator of the integrity of the site and the rapid burial of remains [57]. Also, a fragment of the right shell of the genus *Diplodon* shows evidence of thermal alteration by exposure to heat. Shells affected by this process show changes in original coloration. The absence of anthropic modifications of *Diplodon* sp. at Baño Nuevo site 1, and

the presence of this species of mollusk in freshwater bodies of Central West Patagonia, suggests that the analyzed specimens could have been locally available and likely incorporated as foodstuff.

Finally, two fragments of the species *Aulacomya atra* were identified. This bivalve inhabits rocky or mixed bottoms in coastal marine waters from shallow depths to 40–50 m [71]. *Aulacomya* populations are distributed from southern Brazil to Tierra del Fuego and across the Pacific to Peru [72, 73]. As in the case for the keyhole limpet beads, it was not possible to determine whether the origin of the specimens was from the Atlantic Ocean or the Pacific Ocean.

## Beads and mollusk fragments of Patagonian origin

A total of 11 beads were identified at Baño Nuevo 1. Six were made from shells of *Fissurella* sp., and one possibly from a shell of the Veneridae family. As for the rest, the high degree of modification precluded the identification of the mollusks used as raw material. Beads #4, 5, 6 and 7 are hollowed-out or thinned on both sides and in the central area where the perforation was made. Hollowing and thinning have been documented as means to prepare the surface of considerably thick exoskeletons of mollusks to be drilled [61]. For the oval/elliptical beads manufactured from *Fissurella* sp., the natural perforation or aperture of the shell of this species was used. The recognition of this property has made keyhole limpets to be considered ready-made shell ornaments in several contexts across the globe, including central Chile and in southern Patagonia [74–77]. The beads were obtained from the extraction of the area surrounding the apical orifice, and then, their shape was modified by abrasion and polishing. *Fissurella* sp. inhabit the Atlantic and Pacific oceans, from Brazil to Argentina and from Peru to Chile, respectively [43] (Fig 4). Regarding the beads recovered at Baño Nuevo 1, it was not possible to determine whether keyhole limpets were obtained from the Atlantic or Pacific coast because these mollusks are widely distributed. The degree of modification prevented the recognition of attributes of the natural hole that would have guided more taxonomic precision. Additionally, there are no features of the sculpture or natural coloring of the shells used as raw material for making the beads. The ages associated with the beads are distributed throughout the Holocene, but mostly with the upper levels of SU3 and SU2, which suggests a chronology between 7,900 and 2,900 cal BP for the greater part of the assemblage.

Among the artifacts, a possible preform for an ornamental object was identified. This object was in the initial stages of production, with similar occurrences recorded in various sites in Patagonia [20, 22, 59]. The specimen could have been abandoned for unknown reasons before drilling was completed or likely left at the site by accident. S. Leonardt defined a three-stage operational chain for the manufacture of beads or adornments made from the exoskeletons of mollusks: a) extraction of the base-form; b) modification to produce a preform; and c) perforation of the piece; which allows the identification of products and byproducts of the process of creating adornment-pendant objects [59]. The presence of the preform at Baño Nuevo 1 raises the possibility that part of the manufacture of the shell adornments was carried out at the site. However, this is only one preform, and there is no record of the debris expected in the operational chain, suggesting that it is unlikely a manufacturing context at the site.

## Adornments made from *Megalobulimus* of extra-Patagonian origin

Three pendants were made with exoskeletons of the gastropod *Megalobulimus* sp., Subfamily Megalobuliminae (Strophocheilidae). This snail inhabits tropical and subtropical areas of South America and is the largest land mollusk in the Neotropics, with dimensions in the range between 50 and 160 mm long. It is distributed throughout Colombia, Venezuela, Ecuador, Peru, Bolivia, Paraguay, Uruguay, Brazil, and Argentina, and two species have been recorded

in the Caribbean [48, 49, 78]. Four species are included in the *Megalobulimus* genus: *M. lorentzianus*, *M. musculus*, *M. sanctipauli* and *M. abbreviatus* [49]. The snail has nocturnal habits, remaining buried in the ground or litter during the day and in periods of estivation [49, 50]. The shell of *Megalobulimus* has an oval shape and conical-oval contour; the color can vary by species from dull ivory without luster to pale yellow or brown. Internally, the color can vary from uniform pearly white to amber or pearly brown. The outer lip or peristome has a distinctive pink color, which can be bright and intense or opaque, with a flat-convex morphology, and both the width and thickness vary by species. The sculpture can be thin or thick, with low, regular or irregular ribs depending on the species or portion of the shell [49].

The shells of the mollusks used to create the two complete adornments were identified at the species level: *Megalobulimus lorentzianus*. This species has a low population density and nocturnal and estivation habits and is distributed between southern Bolivia and Argentina, between 20° and 35° S, and between 60° and 65° W [49, 50] (Fig 4). The closest area where *M. lorentzianus* shells could be obtained to manufacture the pendants recovered at Baño Nuevo 1 is at least 1,500 km (NE) away in a straight line.

Ornamental objects made from this genus of mollusks have been recorded in archaeological sites in central and northern Argentina and Chile [51, 52, 79–83]. In Patagonia, in particular, modified *Megalobulimus* fragments have been recorded in diverse archaeological contexts, all of which suggest an extralocal origin (Fig 1). On the northern coast of the Santa Cruz province, two adornments were recovered at the Cueva del Negro site within late Holocene deposits [84]. The Miksa-Knoop collection of the Museo Mario Brozoski in Puerto Deseado city includes an adornment from Colhué Huapi Lake (south center of the Chubut province). Recently, in Colhué Haupi, 13 adornments were recovered at the Oporto 7 site, and *Megalobulimus* was identified as the raw material for another five artifacts from a private collection; most of these artifacts were made from shells of *Megalobulimus lorentzianus*. Also, near this lake, in the Chico River basin, a *Megalobulimus* artifact with peristome was recorded, which is part of the Santiago Pozzi collection housed at the Museo de La Plata [85]. The collections of the Museo Etnográfico JBA in Buenos Aires include an adornment from the San Matías Gulf, Río Negro province [20]. At the Cueva 1 de Pilcaniyeu site, in the southwest region of the Río Negro province, another artifact of this type was recorded, corresponding to a subcircular adornment [20]. At the El Trébol site, in Bariloche city, *Megalobulimus* was identified as the raw material for a bead and a fragment [86]. At the Loma de la Lata site, in the Neuquén province, a subtrapezoidal adornment was recovered [87]. At the Aquihuecó site, in northern Neuquén, 15 beads manufactured from *Megalobulimus* were recorded in association with a primary burial of a subadult individual. At that site, six other similar artifacts were also recovered in surface collections [86]. Recently, a pendant of *Megalobulimus* was recovered from the surface in a site on the southern margin of Viedma lake, in the Santa Cruz province [77]. In this framework, items created through the modification of shells of *Megalobulimus* sp. in Patagonia would have constituted desirable ornamental goods that were part of exchange networks in an extraregional scale [86].

The *Megalobulimus* adornments from Baño Nuevo 1 represent, thus far, the only evidence of such ornamental objects in the eastern fringes of the Andes of Central West Patagonia (Chile). The radiocarbon ages of the layers where these remains were recorded allow us to assign them a chronology of 2,900 cal BP (SU2). This is consistent with the associated chronologies of the other sites in Patagonia where *Megalobulimus* has also been recorded. Therefore, all the evidence considered for Patagonia suggests the existence of a wide network of social interactions between territories and distant groups, which involved the human groups that occupied the Baño Nuevo 1 site, and which implied the transport of items made with *Megalobulimus* during the late Holocene. In Patagonia, several lines of archaeological evidence

demonstrate the increase in the circulation of people and/or objects during the late Holocene [14–17, 88–94]. The presence of exotic goods, such as the *Megalobulimus* adornments from Baño Nuevo 1, would not be related to the regular mobility circuits used for the acquisition of subsistence resources of the hunter-gatherer groups of Central West Patagonia but rather would be part of social contact connections on broader scales, for ritual/ceremonial and/or cultural reasons [13, 95]. These exotic materials would have played an important role in past cultural systems, mainly in the establishment and reaffirmation of social ties between groups and individuals in the context of these long-distance exchange networks, which often involve actions and elements that symbolize ties, materialized through gifts or exchanges, sometimes through ceremonies or rituals [13]. These transactions, involving the exchange of valuable objects, help towards social integration [96, 97]. In this sense, the exchange of goods is a way through which both individuals and societies can build alliances at broad spatial scales, materializing the presence of distant places and people, not available in everyday interactions [97]. Also, these exotic and visually striking objects served as a means of communication and as transmitters of information between populations [20, 62].

Regarding the use of the adornments, two showed wear traces on the upper edge of the perforation and on the closest outer edge, likely as a consequence of prolonged suspension (Fig 5). These traces suggest that the specimens were possibly tied or sewn and not suspended by hanging or as part of necklaces, earrings, or bracelets. Even though the features of use indicate that they could be fastened or sewn, it is not possible to infer whether they were part of

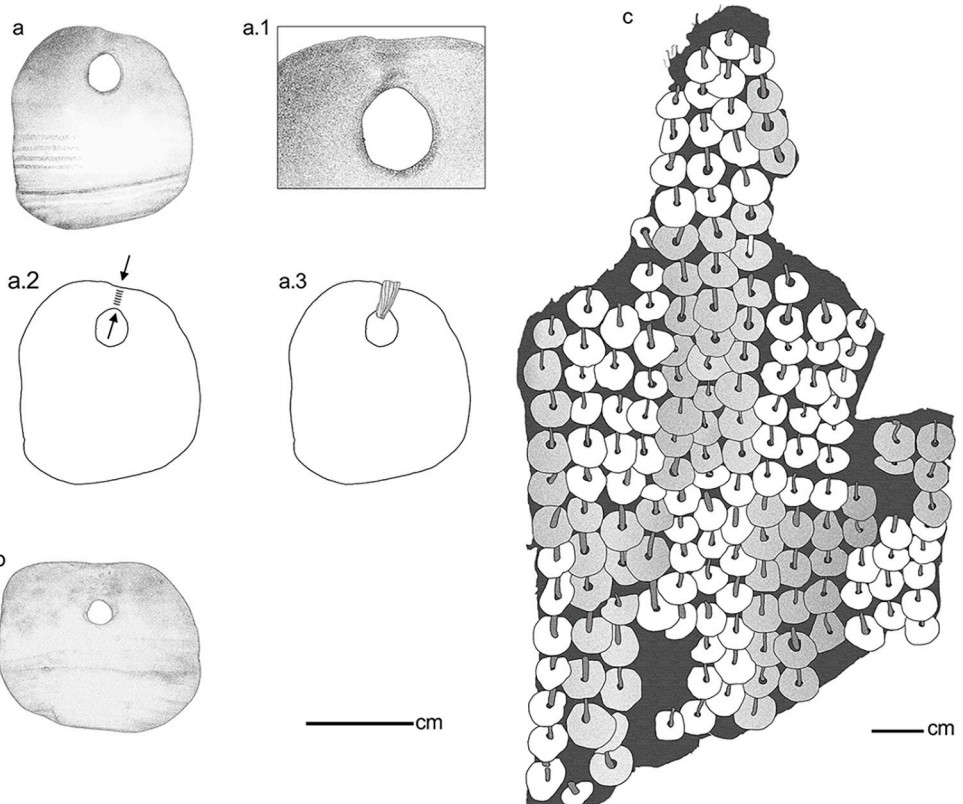

**Fig 5. Depiction of adornments made from *Megalobulimus lorentzianus* from Baño Nuevo 1 and their potential use.** Illustration of pendants: a. specimen #1, a.1. and a.2.: details of the wear and tear; a.3: scheme of the possible means of suspension, b. specimen #2, c. illustration of the leather fragment with sewn shell beads from Sierra Cuadrada [98].

clothing or another individual object. However, there are two antecedents that illustrate the potential use of these objects. In Sierra Cuadrada, in the Chubut province (Argentina), and at a similar latitude to Baño Nuevo 1, a fragment of leather was found with 164 circular shell beads with a sewn leather strip superimposed on each other [98] (Fig 5). The excellent preservation of this archaeological item allows visualizing suspension accomplished by stitching shell ornaments to leather clothing. Additionally, at a site near San Roque in the Córdoba province, north of Patagonia (Argentina), 97 adornments made from *Megalobulimus lorentzianus* were found; the visual and sound characteristics of these items indicate that they were likely multisensory objects used as part of clothing for rituals [99]. In the case of the Baño Nuevo 1 adornments, the available evidence does not allow us to delve into the specific use of these ornaments.

## Conclusion

Although the remains of mollusks found at Baño Nuevo 1 constitute a small archaeomalacological assemblage, they are particularly diverse (good richness). The low frequency is explained by the great distance from the site to the coasts of both oceans, typical sources for mollusks. At Baño Nuevo 1, freshwater, terrestrial and marine mollusks, both from Patagonia and from extra-Patagonian regions, were identified. The presence of these mollusks at a key inland site in Central West Patagonia indicates movements of hunter-gatherer groups on large spatial scales from the beginning of the Holocene onward, although with greater occurrence since the beginning of the middle Holocene. The keyhole limpet shell beads and the *Megalobulimus* adornments indicate human movement across hundreds of kilometers and in various directions.

Objects of exotic origin that are non-utilitarian or unrelated to subsistence, such as personal ornaments manufactured from mollusk shells, have sometimes been defined as "prestige goods" [100]. However, there is no consensus about the meaning and reaches of this concept [101]. In the archaeological and anthropological literature, this term has been often associated with elite goods, status goods, valuable objects or with the wealth of the person or group owning them [12, 102]. In this framework, we consider that the presence of this type of shell ornaments in the context of highly mobile hunter-gatherer groups of Central West Patagonia does not in itself demonstrate the existence of status or social differences, the same as it is not related to obtaining subsistence resources. Rather, the shell ornaments recovered in Baño Nuevo 1 could be evidencing the existence of past extra-Patagonian social interactions, for instance as a result of the exchanges of objects or gifts occurring in the frame of social interactions ranging from visits to marital ties, as expected in broad circulation networks [96, 97]. In this fashion, these ornaments could have played an important role as symbols of social identity, and as materials aiding in the establishment and maintenance of social relationships.

The data reported herein and other archaeological evidence from the Ñirehuao valley and other Andean valleys in Central West Patagonia demonstrate the contact and transport of objects at distances between 350 and 2,500 km, e.g., obsidian of Pampa del Asador and of the Somuncurá Plateau, ceramics with Pampean decorations and metal earrings recorded in funerary structures in the same valley (i.e., BN29) [14, 88, 103]. Although there is evidence for long distance movements from the beginning of the Holocene, such movements increased and extended over time as occupied spaces became more defined and interaction networks gained more relevance for information flows between distant territories of the hunter-gatherers across Patagonia [90]. The limited chronological control for shell-ornament material in Patagonia precludes a more deepened discussion on the transformations of exchange networks over time, while it underscores the importance of the information provided by the Baño Nuevo 1

assemblage. However, the fact that *Megalobulimus* evidence appears in the late Holocene alongside other exotic evidence suggests that the broader Patagonia experienced an increase in the distance, frequency, and rarity of the transported and exchanged items [104]. In this article, shell ornaments materialize the social relationships between people at both, regional and extra-regional scales. As Trubitt has pointed out, sometimes some of these distances are too great to be plausibly explained by seasonal movements and suggest that the shells changed hands as part of a system of exchange between groups, and perhaps, some objects like *Megalobulimus* pendants, because of their rarity, perhaps had an enhanced value that stimulated the circulation of materials and helped to maintain social contacts over large territories [12]. Results herein presented highlight the potential of exotic goods, and particularly that of shell-made artifacts, for understanding how past social networks functioned. Though this type of materials may represent a minor proportion of the archaeological assemblages, understanding them in the frame of broader context, can be key to unveiling aspects of the conception of space, territory, and social ties for hunter-gatherer societies.

## Acknowledgments

We are thankful to Gustavo Saldivia and Juan Pablo Varela (Museo Regional de Aysén) and to Dr. Francisco Mena, the original excavator of Baño Nuevo 1, for their support. Dr. Ariel Beltramino, curator of the Colección Malacológica del Instituto de Biología Subtropical (IBS-Ma; CONICET—Universidad Nacional de Misiones), kindly donated *Megalobulimus* reference samples and commented on the identification of the adornments presented herein. We are indebted to Carola Flores and Daniel Hernández for their support.

## Author Contributions

**Conceptualization:** Heidi Hammond, Leandro Zilio, Amalia Nuevo-Delaunay, César Méndez.

**Data curation:** Leandro Zilio, Amalia Nuevo-Delaunay, César Méndez.

**Formal analysis:** Heidi Hammond, Leandro Zilio, Amalia Nuevo-Delaunay, César Méndez.

**Funding acquisition:** Amalia Nuevo-Delaunay, César Méndez.

**Investigation:** Heidi Hammond, Leandro Zilio, Amalia Nuevo-Delaunay, César Méndez.

**Methodology:** Heidi Hammond, Leandro Zilio, Amalia Nuevo-Delaunay, César Méndez.

**Project administration:** Amalia Nuevo-Delaunay, César Méndez.

**Resources:** Amalia Nuevo-Delaunay, César Méndez.

**Software:** Heidi Hammond, Leandro Zilio, Amalia Nuevo-Delaunay, César Méndez.

**Supervision:** Heidi Hammond, Leandro Zilio, Amalia Nuevo-Delaunay, César Méndez.

**Validation:** Heidi Hammond, Leandro Zilio, Amalia Nuevo-Delaunay, César Méndez.

**Visualization:** Heidi Hammond, Leandro Zilio, Amalia Nuevo-Delaunay, César Méndez.

**Writing – original draft:** Heidi Hammond, Leandro Zilio, Amalia Nuevo-Delaunay, César Méndez.

**Writing – review & editing:** Heidi Hammond, Leandro Zilio, Amalia Nuevo-Delaunay, César Méndez.

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
