## [Decision Letter · Decision Letter 0]

18 Mar 2024

PONE-D-23-41111Middle through late Holocene long-distance transport of exotic shell personal adornments in Central West Patagonia (southern South America). The archaeomalacological assemblage of Baño Nuevo 1PLOS ONE

Dear Dr. Méndez,

Thank you for submitting your manuscript to PLOS ONE. After careful consideration, we feel that it has merit but does not fully meet PLOS ONE’s publication criteria as it currently stands. Therefore, we invite you to submit a revised version of the manuscript that addresses the points raised during the review process.

We look forward to receiving your revised manuscript.

Kind regards,

Miguel Delgado, PhD. Anthropology Division, Faculty of Natural Sciences and Museum, 

National University of La Plata, Argentina

Academic Editor

PLOS ONE

Journal Requirements:

3. Please include a complete copy of PLOS’ questionnaire on inclusivity in global research in your revised manuscript. Our policy for research in this area aims to improve transparency in the reporting of research performed outside of researchers’ own country or community. The policy applies to researchers who have travelled to a different country to conduct research, research with Indigenous populations or their lands, and research on cultural artefacts. The questionnaire can also be requested at the journal’s discretion for any other submissions, even if these conditions are not met.  Please find more information on the policy and a link to download a blank copy of the questionnaire here: https://journals.plos.org/plosone/s/best-practices-in-research-reporting. Please upload a completed version of your questionnaire as Supporting Information when you resubmit your manuscript.

4. In your Methods section, please provide additional information regarding the permits you obtained for the work. Please ensure you have included the full name of the authority that approved the field site access and, if no permits were required, a brief statement explaining why.

5. We note that your Data Availability Statement is currently as follows: All relevant data are within the manuscript

6. We note you have included a table to which you do not refer in the text of your manuscript. Please ensure that you refer to Table 2 in your text; if accepted, production will need this reference to link the reader to the Table.

**Additional Editor Comments:**

As you address the reviewers' comments, we would like for you to attend in particular to: 1) Reviewer #2's concerns regarding the use of ethnographical references to contextualize the inferred ritual and social use of shells in Patagonia; 2) Reviewer #1's suggests a more profound exploration of the anthropological relevance of exchange during the period investigated. 

Reviewers' comments:

Reviewer's Responses to Questions

**Comments to the Author**

1. Is the manuscript technically sound, and do the data support the conclusions?

Reviewer #1: Yes

Reviewer #2: Yes

2. Has the statistical analysis been performed appropriately and rigorously? 

Reviewer #1: N/A

Reviewer #2: N/A

3. Have the authors made all data underlying the findings in their manuscript fully available?

Reviewer #1: Yes

Reviewer #2: Yes

4. Is the manuscript presented in an intelligible fashion and written in standard English?

Reviewer #1: Yes

Reviewer #2: Yes

5. Review Comments to the Author

Reviewer #1: This article delves into a significant archaeological site in northern Patagonia. While the approach is interesting, there is a need for a more profound exploration of the anthropological relevance of exchange during the period in which these artifacts are identified, both at the site and in the broader region under study (refer to suggested bibliography). It is crucial not only to consider the movement of groups over long distances but also to examine the potential exchange of objects. In this context, theoretical aspects related to the significance of the movement of goods should be more thoroughly integrated (please review Lazzari's papers).

Concerning the discussion, it commences abruptly and requires improved organization, demanding greater depth and order. It is essential to start by emphasizing the lines of evidence that demonstrate these regional distributions. A decision must be made regarding whether to present the information by species, and this decision should be explicitly stated. Currently, there are titles for Diplodon and beads, but none for Megabulimus and Fissurella. It would be advantageous to progress by proposing more about possible connections with those other regions. The exploration of regional and theoretical backgrounds, considering the movement and exchange of goods, holds significant potential for better this article. In the central-southern Andean areas, there is considerable advancements, even concerning hunter-gatherer societies.

Reviewer #2: This paper is a well-structured piece of research. A clear problem research is defined, and the methodological apparatus fits this problem. The description of the material is lengthy, detailed and well done. The discussion is well-organized and relevant to understanding the relevance of shells beyond a dietary perspective. In addition, the study of shells from a social and network perspective is appropriate and contributes to expanding the interpretation of this material in archaeology. Combining archaeological data from different Patagonian areas gives the paper a broad perspective. Considering the subject of the paper and the area involved, it will be relevant work to anyone researching social networks, the use of shells, hunter-gatherers, and Patagonian Archaeology.

In my opinion, the paper needs only minor changes to strengthen some points.

On the one hand, ethnographical references about the use of shells in ceremonies or exchange networks could be added. While I recognize the problems related to using ethnographical information to understand archaeological sites from the Middle Holocene, it is necessary to provide a possible contextualization of the ritual and social use of shells in Patagonia.

On the other hand, the authors report the use and circulation of shells in the Middle and Late Holocene; however, they could discuss how these spatial networks could have transformed in this time lapse. As far as I understand, some differences can be identified between both moments in aspects such as mobility and use of space. To deal deeper with this point, it could be useful to insert this evidence in a wider perspective.

Finally, I would suggest reframing the conclusion to increase the paper's impact, highlighting its contribution to the understanding of the social use of shells in prehistory and/or the characteristics of social networks among hunter-gatherers.

6. PLOS authors have the option to publish the peer review history of their article (what does this mean?). If published, this will include your full peer review and any attached files.

Reviewer #1: **Yes: **Catalina Soto Rodríguez

Reviewer #2: No

---

## [Author Response · Author response to Decision Letter 0]

25 Apr 2024

Editorial review List of queries

We have checked the PLOS ONE's style requirements following the provided templates.

2. Please include captions for your Supporting Information files at the end of your manuscript.

The paper does not include any supporting information outside the main body of the manuscript.

3. Please include a complete copy of PLOS’ questionnaire on inclusivity in global research in your revised manuscript. 

We have completed and included the PLOS’ questionnaire on inclusivity in global research.

4. In your Methods section, please provide additional information regarding the permits you obtained for the work. Please ensure you have included the full name of the authority that approved the field site access and, if no permits were required, a brief statement explaining why.

We confirm we have included an explanation for the stage at what the permits were obtained, the person who performed excavations and his approval for laboratory analyses. In the original version we had only indicated the approval by the Museo Regional de Aysén. The authors carried out no field work for this study.

5. We note that your Data Availability Statement is currently as follows: All relevant data are within the manuscript

We confirm that our submission contains all raw data required to replicate the results of your study. All materials used in the article are referenced within the manuscript and these are located at the Museo Regional de Aysén as indicated. Anyone can access the materials following the appropriate Museum protocols. 

6. We note you have included a table to which you do not refer in the text of your manuscript. 

We have now cited Table 2 in the manuscript.

7. Please review your reference list to ensure that it is complete and correct. 

We have revised the reference list and updated all references that changed while the manuscript was under review. Whenever these changes were part of the reviewer’s observations, these were listed in association with the change explained at the end of this letter.

Additionally, the academic editor requested to attend to concerns regarding the use of ethnographical references to contextualize the inferred ritual and social use of shells in Patagonia and to provide a more profound exploration of the anthropological relevance of exchange during the period investigated. Both aspects were thoughtfully considered, and the changes made are indicated in the response to each of the reviewer’s comments.

Reviewer #1 (Catalina Soto Rodríguez): 

Rev. 1 suggested addressing the anthropological relevance of exchange and examining the potential of the exchange of objects in building ties over long distances. In doing so, she suggested some additional references. This comment was addressed by incorporating an additional paragraph in the conclusions section. References suggested by this reviewer resulted beneficial for our paper and were included in the revised version. Papers by Lazzari were included to address the significance of the movement of goods.

Rev. 1 suggested a reorganization of the discussion section, starting by regional distributions and upscaling afterwards. This change has been fully accepted. Also, this reviewer suggested amendments to the subtitles in the discussion section. We changed the subtitles making them more explicit regarding the information included under them. In doing so, we were more explicit with presenting the information by species, as also requested in this review.

Rev 1 also suggested considering the movement and exchange of goods with examples of the central-southern Andes, even among hunter-gatherers, to enhance the discussion. We believe that based on the archaeological context that we are analyzing and its chronologies some of these processes (i.e., emergence of differences in power, prestige, or social values that past groups may have granted to certain elements) cannot be addressed solely by considering the materials that we are analyzing. It is very complex to arrive at this type of interpretation from the assemblage of ornaments from Baño Nuevo 1, and even more difficult if such discussion has not been advanced with other lines of evidence at the scale of the region. Also, the fact that the assemblage was recovered with no association to any of the individuals buried at the site, further limits such potential. Therefore, we included a clarification at the initial part of the results, indicating this last idea. However, as detailed in the original manuscript, there were evident networks of circulation of people and objects in the past through which individuals exchanged various elements, as well as knowledge and information, surely through various modalities as Lazzari suggests, such as for example, exchange or barter, and that do not fall within the sphere of subsistence as we emphasized in the article. In the conclusion section we included a new paragraph regarding this aspect.

Reviewer #2 (Anonymous)

Reviewer #2 has positive comments regarding the structure, definition of the research problem, the utilized methods, the detailed description of the results and the discussion. The reviewer agrees with the relevance, considers that the perspective is appropriate and states that the paper contributes to expanding the interpretation of this material in Patagonian archaeology. 

Rev. 2 also indicated the need of strengthening ethnographical references about the use of shells for ceremonies or exchange networks to contextualize ritual and social uses of shells. However, there is no available ethnographic knowledge regarding such uses for Tehuelche/Aonikenk groups. Though this kind of data may be available for other parts, including coastal societies elsewhere in Patagonia (e.g., groups in Tierra del Fuego or other archipelagic settings), these groups are remarkably different from those who inhabited Central South continental Patagonia, particularly in terms of mobility, the use of territory, proximity and use of coastal resources, and social organization. These are all key implications of our record. This situation is enhanced by the lack of any ethnographic records in Central West Patagonia (the region where Baño Nuevo 1 is inserted). We have included a few additional sentences and references in the introduction to address this point.

Also, this rev. suggested discussing how the spatial networks transformed over time lapse inserting this evidence in a wider perspective. In the conclusion section, we included an additional paragraph and two insertions that discusses concepts proposed by the reviewers and shows our position with regarding to these topics. With this paragraph we respond to several questions raised by both reviewers. In this way, we believe that the conclusions are deepened as requested. Such changes involved including new references that have enhanced the conclusions of the article.

Finally, Rev. 2 suggested reframing the conclusion to increase the paper's impact, highlighting its contribution to the understanding of the social use of shells among hunter-gatherers. We included an additional sentence at the end of the article stating that claim.

Additional changes introduced by the authors.

To update the information in the article, we included:

1. A change in the corresponding author’s, César Méndez, affiliation following his change to the Estudios Aplicados unit of the Pontificia Universidad Católica de Chile. 

2. A new version of Figure 1 including additional sites relevant to this study that were published while the manuscript was under review.

3. The appropriate reference for the above (Leonardt et al. 2024).

4. A new version of Figure 4 with a minor amendment.

5. An additional reference to the archaeology of Baño Nuevo 1 that was published while the manuscript was under review (Méndez et al. 2024).

6. An update to Hammond et al. 2024’s reference definitive form, originally under review, while the manuscript was under review.

---

## [Editor Report · Decision Letter 1]

14 May 2024

Middle through late Holocene long-distance transport of exotic shell personal adornments in Central West Patagonia (southern South America). The archaeomalacological assemblage of Baño Nuevo 1

PONE-D-23-41111R1

Dear Dr. Méndez,

We’re pleased to inform you that your manuscript has been judged scientifically suitable for publication and will be formally accepted for publication once it meets all outstanding technical requirements.

Kind regards,

Miguel Delgado, PhD. Anthropology Division, Faculty of Natural Sciences and Museum, 

National University of La Plata, Argentina.

Academic Editor

PLOS ONE
---

## [Editor Report · Acceptance letter]

16 May 2024

PONE-D-23-41111R1 

PLOS ONE

Dear Dr. Méndez, 

I'm pleased to inform you that your manuscript has been deemed suitable for publication in PLOS ONE. Congratulations! Your manuscript is now being handed over to our production team.

Kind regards, 

on behalf of

Dr. Miguel Delgado 

Academic Editor

PLOS ONE